# Nitrogen Dynamics Following Incorporation of 3-Year Old Grassland Set-Asides in the Fraser River Delta of British Columbia

**Khalil Walji [1,2], Maja Krzic [1,3]** ⓘ**, Drew Bondar [4] and Sean M. Smukler [1,*]** ⓘ

[1]   Soil Science Program, Faculty of Land and Food Systems, University of British Columbia, 2357 Main Mall, Vancouver, BC V6T 1Z4, Canada; khalilwalji@gmail.com (K.W.); maja.krzic@ubc.ca (M.K.)

[2]   Food and Agriculture Organization of the United Nations (UN-FAO), 00153 Rome, Italy

[3]   Department of Forest and Conservation Sciences, Faculty of Forestry, University of British Columbia, 2424 Main Mall, Vancouver, BC V6T 1Z4, Canada

[4]   Delta Farmland and Wildlife Trust, 205-4882 Delta Street, Delta, BC V4K 2T8, Canada; drew@deltafarmland.ca

\*   Correspondence: sean.smukler@ubc.ca; Tel.: +604-822-2795

**Abstract:** Short-term grassland set-asides (GLSA) have been incorporated into intensive annual crop rotations to improve soil quality. The legacy of the GLSA to subsequent annual crops, however, is not well understood. The objective of this study was to determine the impacts of 3-year-old GLSA on nitrogen (N) dynamics and the yield of the subsequent cash crop. A regional analysis was conducted over two years, utilizing eight production fields transitioning from GLSA, paired with fields in continuous annual crop rotation (ACR) with matching management. A controlled plot-scale experiment was also conducted on a single 3-year-old GLSA, comparing fertilizer types, rates, and timing of incorporation. In each experiment, soils were sampled every 10–14 days for ammonium ($NH_4^+$-N) and nitrate ($NO_3^-$-N), along with ion probes, installed near the rooting zone to track plant available nitrogen (PAN) throughout the season. The results from the regional analysis are confounding, in 2015 showing that GLSA supplied an additional 18 kg PAN ha$^{-1}$ compared to ACR but showed no PAN benefits in 2016. The controlled plot-scale experiment highlighted the importance of fertilizer type to subsequent PAN, showing synthetic treatments consistently supplied more PAN than organic. The results from this study suggest that 3-year-old GLSAs can potentially improve PAN to subsequent crops depending on how they are managed.

**Keywords:** plant available nitrogen; vegetable yields; nitrate; ammonium; immobilization; mineralization

## 1. Introduction

Grassland set-asides (GLSA), also known as grassland-arable rotations and grass-leys, are a management scheme that replaces crop production on arable land with perennial grasses for various economic and ecological purposes [1]. Between 1988 and 2008, the European Union utilized a set-aside program as an economic tool to control the supply of cereal crops [2], while the United States in 1985 established a reserve program, comprised of millions of hectares of GLSA to reduce soil erosion and to promote wild bird habitat [3]. In Canada, three GLSA programs have been utilized since 1988, to control soil erosion as well as create bird habitat. Of these three programs, only one remains operational today—the GLSA stewardship program offered by the Delta Farmland and Wildlife Trust (DF&WT), a non-governmental organization in the Fraser River Delta of British Columbia (BC). This program provides cost-share payments to farmers for taking active agricultural land out of production and seeding it with a mix of grasses and legumes for a period of one to four years [4]. The main goals of

the GLSA program are to provide wildlife bird habitat in a critical nesting area (the Pacific flyway) and enhance soil organic matter (SOM) and structure, but farmers also enroll their fields to transition to organic production or to include these short-term GLSAs into annual crop rotations (ACR) [5].

While studies carried out on long-term (i.e., >10 years) GLSAs have reported increases in SOM and plant available nitrogen (PAN) [1,6,7], the effects of short-term GLSAs, such as those used in the Fraser River delta of BC, on nitrogen (N) dynamics and PAN, are unclear. Incorporating the biomass from GLSAs has been shown to increase rates of N mineralization and PAN [8,9], and when combined with synthetic fertilizer applications, have been shown to increase N mineralization [10], improve crop productivity and reduce long-term losses of SOM [11], in some cases even improving SOM formation [12]. At the same time, it is well-known that incorporating organic materials (e.g., GLSA biomass) with high (>25) C:N ratios can temporarily immobilize N [13,14]. If immobilization of N occurs during key growth stages of the cash crop that is subsequently grown after GLSA incorporation, the cash crop yields may be negatively impacted and could result in unused PAN post-harvest [8]. Given the timing of N release has not been well documented in the region, farmers in the Fraser River delta typically apply N fertilizers when incorporating GLSA biomass to compensate for a perceived reduction in PAN or the asynchrony between PAN release and the uptake of the cash crop. While the addition of N fertilizers may help compensate for potential immobilization, over-application of N represents an economic loss to farmers and unused PAN, particularly in regions such as the Fraser River delta that have heavy wintertime precipitation, can become a pollutant to surface and groundwater via runoff and leaching.

There are a number of other factors that likely contribute to the amount and timing of PAN release beyond the C:N ratio and amount of GLSA biomass incorporated. The timing of GLSA incorporation and subsequent sowing of the cash crop can play an important role in the N dynamics [14]. In the Fraser River delta region, where there are cool wet winters, spring farm operations are typically constrained by soil moisture, and farmers generally use the first opportunity of dry weather to prepare their fields. However, the later the incorporation of the GLSA, the warmer the temperature is likely to be. The timing of GLSA mowing and incorporation in combination with soil moisture and temperature is likely an important determinant of decomposition rates and N mineralization and thus the amount and timing of PAN release. It is also likely that the amount as well as the type of fertilizer used in combination with incorporated GLSAs will influence decomposition rates and N mineralization [14]. Fertilization using composts or manures for example, will not only provide additional C along with N applications, thus modifying the overall C:N ratio of the system, they will also provide substantial quantities of microbial populations [15]. Given that one of the goals of farmers adopting GLSA is to transition to organic production, it is important to understand how fertilization based on organic inputs will impact the N dynamics following GLSA.

The overarching objective of this study was to determine the impacts of 3-year-old GLSA (a typical duration of GLSAs in the Fraser River delta of BC) on N dynamics and yield of the subsequent cash crop. To do so, we conducted an observational experiment to track PAN following GLSA incorporation under a range of production conditions across a number of operational farms over two seasons, and a complementary controlled plot experiment where variation in conditions was minimized over one season. Our specific objectives were to: (i) compare the timing and quantity of PAN and crop yields on operational farm fields that were in GLSA for the last three years to those in continuous annual crop rotations (ACR) over two growing seasons; and (ii) evaluate how post GLSA N mineralization changes with the quantity, timing, and type of fertilizer in a controlled plot scale experiment. Our first hypothesis was that the incorporation of a 3-year-old GLSA with the same fertilizer application rate as ACR would result in increased PAN in the growing season following incorporation, leaving nitrates in the soil at the end of the season, increasing the risk of leaching losses, without improving crop yields. Our second hypothesis was that delayed incorporation of GLSA would increase the quantity of PAN given warmer temperatures later in the season. Our third hypothesis was that the incorporation of 3-year-old GLSA with a high rate of fertilizer (200 kg N ha$^{-1}$) would hasten the release of N and

increase cumulative PAN compared to the typical application rate (100 kg N ha$^{-1}$). Our final hypothesis was that organic fertilizer in combination with incorporated GLSA would result in slower N release, but equivalent cumulative PAN as synthetic fertilizer.

## 2. Materials and Methods

### 2.1. Study Sites

#### 2.1.1. On-Farm Regional Assessment

We conducted the observational experiment in 2015 and 2016 across eight paired operational fields that had either been in continuous ACR or GLSA for the last three years. All study fields were located in the lower Fraser River delta in the municipality of Delta, BC (49.0847° N, 123.0586° W), about 30 km south of the city of Vancouver (Figure A1). The study fields were located on silty loam to silty clay loam Humic Gleysols [16], Mollic Gleysols in FAO soil classification [17]. This region is characterized by a humid, temperate climate with a mean annual temperature of 11.1 °C and a mean annual precipitation of 1189 mm [18], with 80% of rainfall occurring between October and April. The average temperature in 2015 was 11.4 °C, with an annual precipitation of 1141 mm [18]. The 2015 temperature was 9% higher and precipitation 4% lower than the 30-year historical average. In 2016, the average temperature at sites was 11.2 °C with a reported annual precipitation of 1315 mm, 7% and 10% higher, respectively than the 30-year historical average (Figure A2).

The GLSA fields were seeded with a mixture of grass and legumes developed by the DF&WT [4], comprised by seed weight of 25% orchard grass (*Dactylis glomerata* L.), 28% tall fescue (*Festuca arundinacea* Schreb.), 30% short fescue (*Festuca rubra* subsp. *commutata* Gaudin and *F. rubra* subsp. *rubra* L.), 15% Timothy grass (*Phleum pratense* L.), and 2% red clover (*Trifolium pratense* L.). The species and proportions which comprise the seed mix were selected through previous studies with the goal of improving soil degradation, commonly caused by the intensive tillage practices used for potato production in this region [19]. The GLSA mix is also comprised of plant species that are salt tolerant and resist browsing by the large populations of migrating waterfowl that visit the region every year while meeting the other objectives of the DF&WT program listed above. Biomass of the GLSA was assessed prior to mowing, while all other observations were done following GLSA cessation once fields were prepped and planted with a vegetable crop (either beans, potatoes or broccoli). In both 2015 and 2016, GLSA biomass was incorporated 11–59 days before crop planting, based on the weather conditions and each farmer's schedule. Incorporation and field preparation of GLSA and ACR fields included mowing, pulvi mulching, subsoiling and disking numerous times depending on the site. Farmers tried to keep the management of each site as similar as possible but nutrient management varied in between sites and included both organic and synthetic fertilizer being applied at various rates [20].

#### 2.1.2. Controlled Plot Scale Experiment

The controlled plot-scale experiment was conducted from April to August of 2016 on a single 3-year-old GLSA field, seeded in the same grass mix described above to compare fertilizer type, rates, and timing on soil N dynamics. An area of ~100 m$^2$ of the field was utilized for this experiment and the rest was left in set-aside. The site was on a silty clay loam Humic Gleysol [16], Mollic Gleysols in FAO soil classification [17]. The 2016 climate record of a nearby weather station indicated an average temperature of 11.2 °C and annual precipitation of 1316 mm at the site, 9% and 4% higher than the 30-year historical average, respectively (Figure A2). This experiment was laid out as a randomized complete block design with four blocks and following five treatments:

1    High fertilizer rate of 200 kg N ha$^{-1}$ and GLSA aboveground biomass (AGB) incorporated (High),
2    Typical fertilizer rate of 100 kg N ha$^{-1}$ and AGB incorporated (Typical),
3    Late incorporation of ABG, 14 days after other treatments, with application of 100 kg N ha$^{-1}$ (Late)

4　　　Composted chicken manure applied at 100 kg N ha$^{-1}$ with AGB incorporation (Organic),

5　　　No fertilizer applied and no AGB incorporated (Control).

In all treatments with synthetic fertilizer application, the all-purpose blend of ammonium sulfate $(NH_4)_2SO_4$ contained 13% N, 16% phosphate, 10% K, and 11% S. The grass in the entire experimental site was mowed using a hand held four stroke brush cutter, and incorporated at tilling depth of 30 cm using a Mid-Tine Rotor-Tiller. Three passes were made to ensure complete incorporation of residue to simulate typical field preparation done by farmers in this region.

## 2.2. Sampling

For the regional analysis of operational farms, a baseline set of soil samples were taken at the beginning of 2015 and 2016 growing seasons to determine a suite of soil properties, with subsequent sampling to determine PAN done every two to three weeks in 2015 and every four weeks in 2016, starting after crop planting and continuing until crop harvest. Samples were taken at 0–15 cm and 15–30 cm depths from four randomly located subplots. Paired fields were always sampled on the same day. While these soil samples were important for quantifying area-based values of PAN, they only provided a snapshot in time. To ensure that we did not miss important mineralization events and compare cumulative PAN, we also deployed ion exchange resin probes. Ion exchange membranes were used over the growing season at 2 to 4 week intervals. Probes with a surface area of 10 cm$^2$, made from ion resin membranes which adsorb cations and anions, were installed vertically into the top 15 cm of the soil profile. Two probe pairs were placed in each subplot.

In the controlled plot-scale experiment, soil samples were taken seven times during the 2016 growing season, starting at seven days after incorporation (DAI) and continued in 7 day intervals until 21 DAI, at which point sampling frequency was extended to every 3 weeks until 131 DAI. At each sampling time, two sub-samples per plot were taken at 0–15 cm and 15–30 cm depths and mixed to create a composite sample for each depth. One ion exchange probe was used in each plot and removed at 2 to 4 week intervals.

Prior to GLSA incorporation, AGB was harvested from 1 m × 1 m plots at 12 randomly selected locations of the regional analysis sites. In the controlled plot-scale experiment, AGB from only the control plots were mowed and collected before tilling. After weighing wet samples, biomass was dried at 60 °C for five days to determine dry matter content (DM). Composite sub-samples of AGB from each plot were then ball milled and analyzed for carbon (C) and nitrogen (N) using high-temperature flash combustion [21] with an Elemental Vario El Cube elemental analyzer (Elementar Analysensysteme GmbH, Hanau, Germany). Concentrations of C and N in AGB were multiplied by the dry mass at harvest to determine the additional C (AC) and additional N (AN) added as incorporated residue.

At the end of 2015 and 2016 growing season, 12 subplots of 1 m × 1 m were harvested to estimate crop yield in the regional analysis, while no crops were grown in the controlled plot-scale experiment. Crop sub-samples were freeze dried at −55 °C for four days using a Labconco Model Freeze drier (Labconco, Kansas City, MO, USA) to determine DM content.

A relative yield was calculated for each field pair using Equation (1) to compare yields between GLSA and ACR, and to account for the variety of crops grown across the farms included in this study.

$$RY = \frac{\sum_{n=1}^{4} \frac{SY}{MY}}{4} \tag{1}$$

where the *RY* is the relative yield (%) for each field, *MY* is the maximum yield observed for a sub-plot at each site (each GLSA and ACR pair), and *SY* is the sub-plot yield.

## 2.3. Laboratory Analysis

Baseline soil samples were air dried for one week, crushed using a wooden rolling pin to pass through a 2 mm sieve, and percentage coarse fraction was calculated for each sample. Subsamples

of 100-g were taken from 25% of the samples and sent to the Technical Service Laboratory of British Columbia Ministry of Environment and analyzed for total soil C (TC) and total soil N (TN) by the dry combustion method [22] and for soil texture using the hydrometer method [23]. The same subset of samples was analyzed at the University of British Columbia (UBC) for pH and electrical conductivity (EC) in distilled water at a suspension ratio of 1:2 and for exchangeable pH in 0.4 $M$ CaCl$_2$ at a suspension ratio of 1:20 [24]. All soils were then analyzed using Fourier-transformed mid-infrared spectroscopy (FT-MIR) with a Tensor 37 HTS-XT spectrometer (Bruker Optics, Billerica, MA, USA). Partial least squares regression using Quant package in OPUS 7.2 (Bruker Optik GmbH, Ettlingen, Germany, ) was used to develop predictions of soil properties from the relationship between FT-MIR spectra and laboratory results. These models were then used to predict soil properties for the remaining 75% analyzed only by FT-MIR [25].

Soil samples for PAN determination were transported in coolers and extracted using 2$M$ KCl, in a ratio of 10 g of soil to 25 mL of KCl and frozen until analysis, which was done colorimetrically using a 99 well microplate absorbance reader (Biorad iMark, Hercules, CA, USA) following the method of Doane and Horwath [26]. A 20 g soil subsample was oven dried at 105 °C until reaching a stable weight to determine gravimetric water content [27]. Concentrations of elements were converted to area values (kg ha$^{-1}$) using soil bulk density calculated with a pedo-transfer function in Equation (2) [28].

$$BD = 1.72 - 0.294(TC)^{0.5} \tag{2}$$

where BD is the bulk density at each depth and TC is the concentration of total soil C.

Initially probes purchased from Western Ag Innovations Inc. (Saskatoon, Canada) were used three times in the 2015 growing season and were sent to the Western Ag laboratory for analysis after which ion exchange probes constructed from membrane sheets purchased from GE Power and Water (Ion probes) were utilized for the remainder of the study. Ion probes were brushed clean of soil, extracted with 2$M$ KCl, and cumulative NH$_4^+$-N and NO$_3^-$-N supply were determined on the extract colorimetrically as described above. The correlation between Western Ag probes and GE ion probes was assessed ($r^2 = 0.7$) and then used as an equivalent method. Western Ag probes were converted to the same scale as GE ion probes and cumulative PAN was reported over the season. Values of adsorbed PAN were converted from μg ml$^{-1}$ to μg 10 cm$^2$ using the following Equation (3) [29].

$$\mu g \; PAN \; 10 \; cm^2 \; = \; \frac{\frac{PAN \; \mu g}{ml} \; \times \; \frac{extract \; ml}{probe} \; \times \; \frac{n_{probe}}{tube}}{20 \; cm^{-1}} \; \times CF \tag{3}$$

where *PAN* is either NO$_3^-$-N or NH$_4^+$-N, extract is the volume of extraction per probe, $n_{probe}$ is the number of probes extracted together per individual falcon tube and *CF* is a conversion factor of 10.

In the controlled plot-scale experiment, short term aerobic incubations were run at four times during the 2016 growing season (i.e., at 7, 22, 76 and 131 DAI) to assess potentially mineralizable N (PMN). Subsamples of 8 g collected at 0–15 and 15–30 cm depths were put into specimen cups and brought to 50% water holding capacity. Samples were left to incubate in a dark cupboard box in the lab (at temperature of 21–25 °C) for 14 days and every 3–4 days' moisture was adjusted to keep constant. After 14 days, samples were extracted with 2$M$ KCl and frozen until analysis, which was done colorimetrically as described above. The PMN rate was calculated using Equation (4) [30].

$$PMN = \frac{N_i - N_f}{incubation \; days} \tag{4}$$

where $N_i$ is the initial mineral N extraction and $N_f$ is the final mineral N extraction over the total of the incubation days, which in this case was 14.

### 2.4. Statistical Analyses

In the regional analysis of operational farms, baseline soil properties were tested using a linear model for differences between treatments and years. An ANOVA was used to determine significant effects with treatment, year, and treatment × year interaction as fixed effects and block as a random effect. Significant differences between treatments and year were differentiated utilizing Tukey's honestly significant difference test.

To determine differences in PAN and cumulative N supply between GLSA and ACR, a linear mixed effects (LME) model was used. The LME model allowed for correlation between temporally repeated measurements collected over the season utilizing an autocorrelation structure and also allowed for correlation between samples from the same spatial locations. Both depths and fields seasons were analyzed separately as weather, management, and patterns greatly differed. In cases where the data were non-normal they were transformed using a log base 10 transformation to meet the assumption of normality and homoscedasticity. Residual soil N was also analyzed using a LME model; however, the model excluded any time correlation as analyses was run solely on the final sampling point. A Type 3 ANOVA was used to test for significant differences ($p < 0.05$) between main effects (GLSA and ACR). A correlation matrix was run using Pearson's correlation and utilizing the Corrgram package [31] between various soil properties, sampled and climate variables.

For the controlled plot-scale experiment, differences in PAN and cumulative N supply between treatments effects (High, Typical, Late, Organic, Or Control) a linear mixed effects (LME) model was used with treatment, date, and treatment × date interaction as fixed effects and block as a random effect. Depths and sample dates were analyzed separately as there was a significant interaction between sample date and treatment. Non-normal data were log10 transformed to meet assumptions of normality and homoscedasticity. Models which did not meet these assumptions even after transformation were analyzed using the non-parametric Wilcox test function. A Type 3 ANOVA was again used followed by Tukey's honestly significant difference test. All analyses were computed using R Version 3.2.2 [32] and the NLME package version 3.1 [33].

## 3. Results

### 3.1. On-Farm Regional Analysis of Grassland Set-Asides

The pattern of both the timing and overall average PAN between GLSA and ACR differed in the 2015 and 2016 growing seasons (Figures 1 and 2). In 2015, average $NO_3^--N$ in the 0–15 cm depth was significantly ($p < 0.05$) higher at sampling time one and again at sampling time four and six (Figure 1A). The greatest difference was observed at sample four with the GLSA supplying 29 kg ha$^{-1}$ more than control fields. Sampling time six, at the end of the 2015 season, was indicative of residual soil N and it showed that GLSA left 8 kg ha$^{-1}$ more $NO_3^--N$ in the soil than ACR. In 2015, the overall seasonal average $NO_3^--N$ was 26% and 15% greater ($p < 0.05$) in GLSAs than ACR at 0–15 cm and 15–30 cm depths, respectively (Figure 1B). In contrast, during the 2016 growing season, overall seasonal average $NO_3^--N$ was 41% and 25% greater ($p < 0.001$) in ACR than GLSAs at the 0–15 cm and 15–30 cm depth, respectively (Figure 1F,H).

The timing of $NH_4^+-N$ supply during the 2015 season followed a similar trend to $NO_3^--N$. At sampling times one, four, and five, $NH_4^+-N$ at the 0–15 cm depth was significantly higher ($p < 0.05$) in GLSA than in ACR (Figure 2A), and the overall seasonal average $NH_4^+-N$ was 20% higher ($p < 0.01$) in GLSAs than in ACR fields (Figure 2B). At the 15–30 cm depth, $NH_4^+-N$ was only significantly higher in GLSA at sampling time three (Figure 2C) and the overall seasonal average $NH_4^+-N$ was not significantly different between GLSA and ACR (Figure 2D). In 2016, there were no significant differences in $NH_4^+-N$ either by sampling time or overall average at either depth (Figure 2E–H).

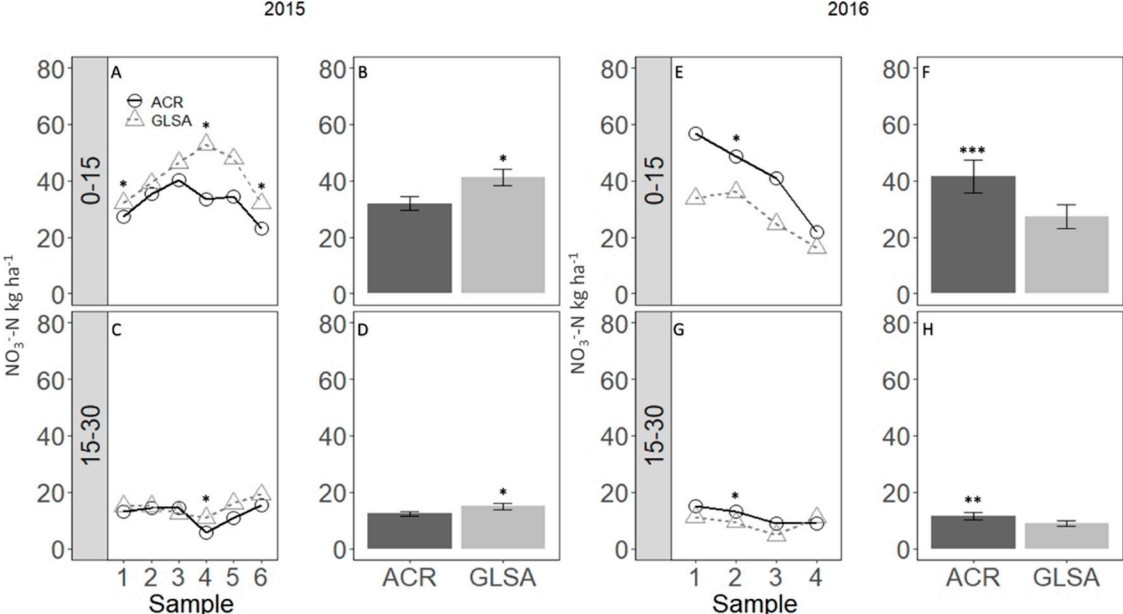

**Figure 1.** Average soil nitrate ($NO_3^-$-N) in 2015 from fields with annual crop rotations (ACR) and following a 3-year grassland set-aside (GLSA) at the 0–15 cm depth by sampling time (**A**) and seasonal average (**B**) and at the 15–30 cm depth by sampling time (**C**) and seasonal average (**D**); average soil nitrate ($NO_3^-$-N) in 2016 from fields with ACR and following a 3-year GLSA at the 0–15 cm depth by sampling time (**E**) and seasonal average (**F**) and at the 15–30 cm depth by sampling time (**G**) and seasonal average (**H**). Error bars represent one standard error of the mean (n = 4). Significant differences are indicated by * ($p < 0.05$), ** ($p < 0.01$), and *** ($p < 0.001$).

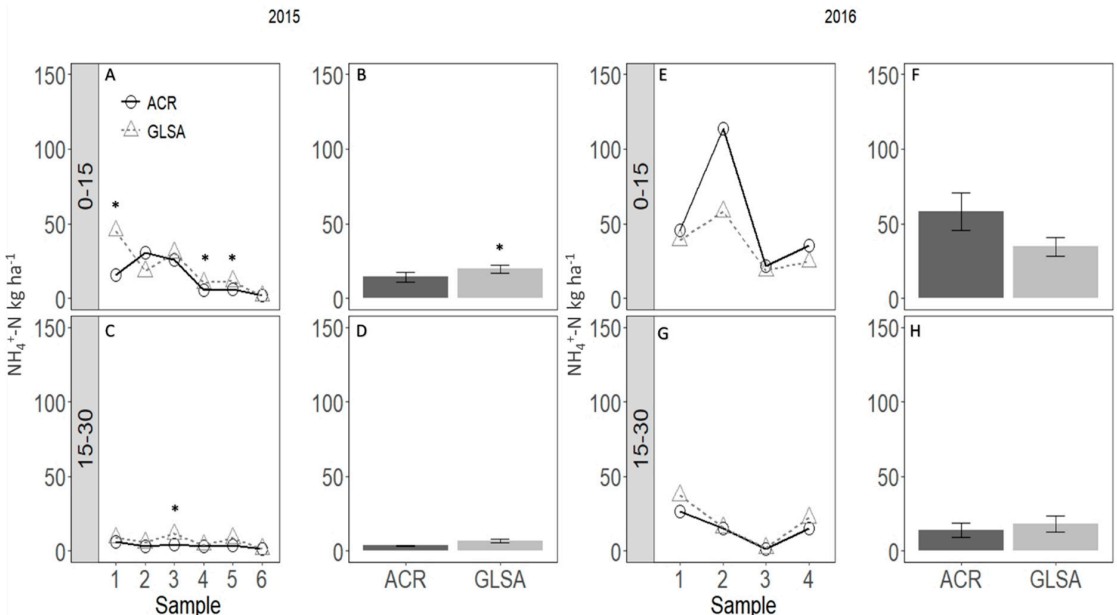

**Figure 2.** Average soil ammonium ($NH_4^+$-N) in 2015 from fields with annual crop rotations (ACR) and following a 3-year grassland set-aside (GLSA) at the 0–15 cm depth by sampling time (**A**) and seasonal average (**B**) and at the 15–30 cm depth by sampling time (**C**) and seasonal average (**D**); average soil ammonium ($NH_4^+$-N) in 2016 from fields with ACR and following a 3-year GLSA at the 0–15 cm depth by sampling time (**E**) and seasonal average (**F**) and at the 15–30 cm depth by sampling time (**G**) and seasonal average (**H**). Error bars represent one standard error of the mean (n = 4). Significant differences are indicated by * ($p < 0.05$).

In the 2015 growing season (that was warmer and drier than typical), cumulative $NO_3^--N$ was not significantly different until sampling time five ($p < 0.001$) when GLSA supplied 67% more $NO_3^--N$ than ACR (Figure 3A). In contrast, during the 2016 growing season (that was warmer and wetter than typical), ACR sites supplied 80% ($p < 0.01$) more $NO_3^--N$ by the end of the season (Figure 3B). There were no significant differences observed in cumulative $NH_4^+-N$ during the 2015 season, while by the end of the 2016 season the ACR supplied three times as much $NH_4^+-N$ ($p < 0.01$) than the GLSA (Figure 3C,D).

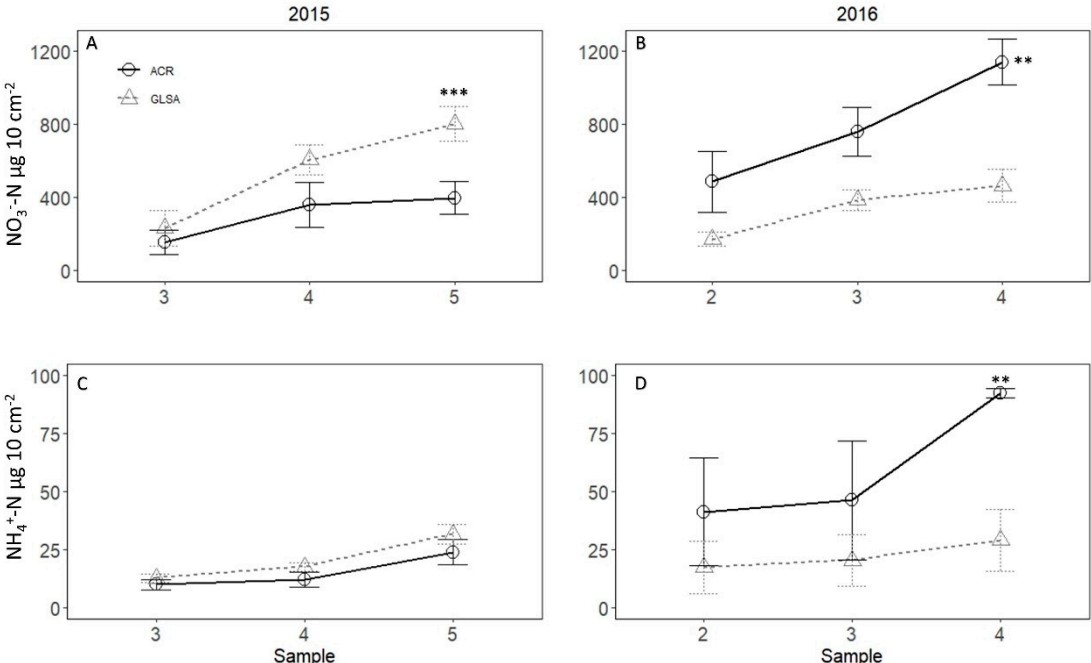

**Figure 3.** Average cumulative nitrate ($NO_3^--N$) supply from fields with annual crop rotations (ACR) and following a 3-year grassland set-aside (GLSA) during the 2015 growing season (**A**); during the 2016 growing season (**B**); cumulative ammonium ($NH_4^+-N$) supply during the 2015 growing season (**C**); and during the 2016 season (**D**). Error bars represent standard error of the mean (n = 4). Significant differences are indicated by ** ($p < 0.01$), and *** ($p < 0.001$).

In 2015, relative yields for potatoes and beans varied very little across the region and were not significantly different between GLSA and ACR. Relative yields for GLSA were 78 ± 1.5% (mean ± SE), and for the ACR they were 85 ± 4.7%. In 2016, relative yields for potatoes and broccoli again were not significantly different but the variation observed was much higher due to pest damage in one GLSA field and sampling problems which resulted in the loss of one replicate pair. As a result, the GLSA relative yield was 63 ± 22%, while the ACR was much less variable with a relative yield of 83 ± 6.8%.

### 3.2. Controlled Plot-Scale Experiment

To supplement findings obtained during 2015 in the regional on-farm analysis, a controlled plot-scale experiment was established in 2016, representing some of the management practices typically used by farmers in this region. Similar to the regional analysis, the incorporated AGB of GLSA was largely comprised of grasses with an average C concentration of 43% and average N concentration of 2%, the average C:N ratio was 28.5, with AGB that on average supplied 48 kg ha$^{-1}$ of AN to the soil (data not shown). Baseline soil properties at the 0–15 cm depth were similar in all blocks; with 2.5% TC, 0.24% TN, pH 4.94, and EC of 0.98 ds m$^{-1}$.

During the 2016 growing season, PAN at the 0–15 cm depth differed among treatments (Figure 4A,B), but not at 15–30 cm depth (data not shown). The pattern of $NO_3$ release varied

by treatments, but the clear peak of $NO_3^-$-N availability was observed on 22 June 2016 (Figure 4). At that date, the High and Typical treatments supplied 56 and 73 kg $NO_3^-$-N ha$^{-1}$ more, respectively, than the Control treatment. The Late treatment, where AGB was incorporated two weeks later than on other treatments, had significantly ($p < 0.05$) lower peak compared to the other synthetic fertilizer treatments, but was not different from the Organic and Control treatments.

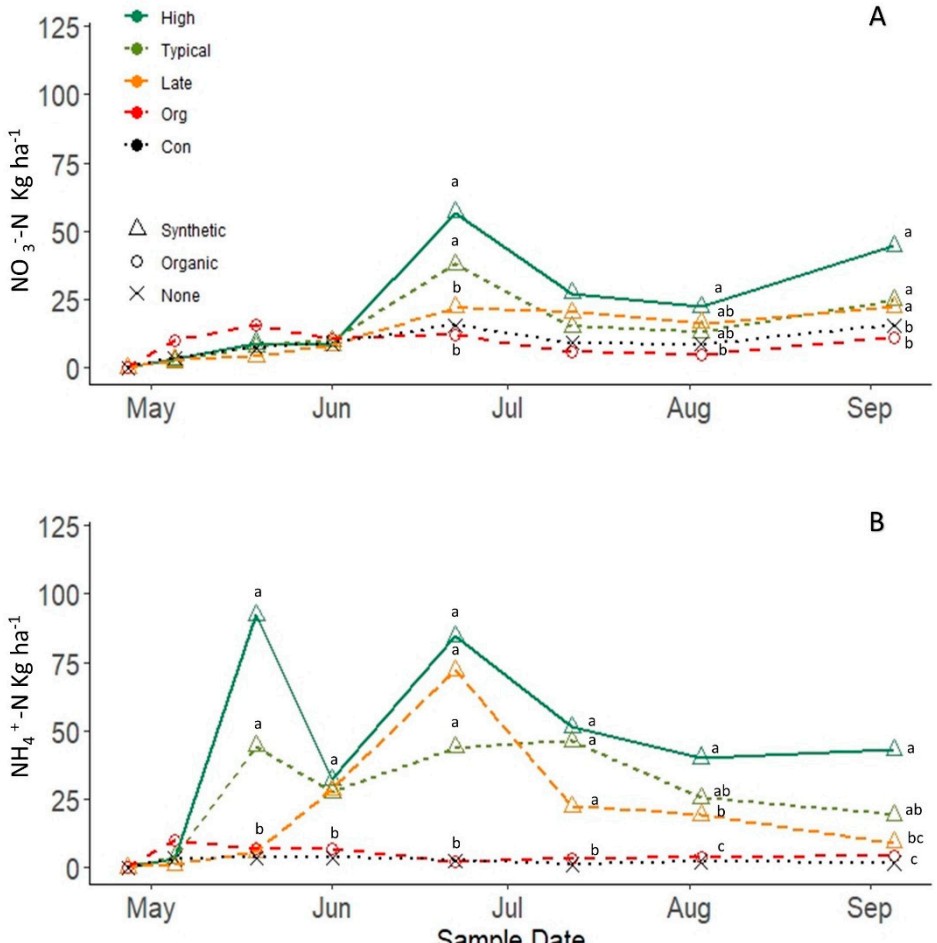

**Figure 4.** Average soil nitrate ($NO_3^-$-N) (**A**) and average soil ammonium ($NH_4^+$-N) (**B**) at the 0–15 cm depth during the 2016 growing season on the controlled plot-scale experiment. Triangles indicate synthetic fertilizer; open circles are organic fertilizers and X symbols no amendments. Results are means of four replicates and different letters indicate significant differences ($p < 0.05$).

Since High, Typical and Late treatments all received $NH_4$-based fertilizer, it is not surprising that the peak of $NH_4^+$-N was observed earlier in the growing season than $NO_3^-$-N (Figure 4B). In samples taken on May 18th (21 DAI), the $NH_4^+$-N supply of High and Typical treatments was 25 and 11 times greater, respectively, relative to the Control and differed significantly ($p < 0.05$) from all other treatments but not from one another. By 1st June (35 DAI), the $NH_4^+$-N in the High and Typical treatments dropped substantially, while the Late treatment increased to match levels in High and Typical treatments, all three then supplying close to ten times more ($p < 0.05$) $NH_4^+$-N than the Control and four times as much as Organic treatments.

At the final sampling time on 5th September (131 DAI), $NO_3^-$-N supply in all treatments increased slightly, and High and Typical treatments had significantly ($p < 0.05$) higher $NO_3^-$-N than Control and Organic treatments. At the final sampling time, the high treatment also supplied 25 times more $NH_4^+$-N, than Control and had significantly more $NH_4^+$-N ($p < 0.05$) than all other treatments (Figure 4B) except

for Typical, which supplied 11 times more $NH_4^+$-N than control and differed significantly from both the Organic and Control treatments ($p < 0.05$), but not from the Late treatment. The Organic treatment was not significantly different from the Control at any sampling date throughout the experiment in either $NO_3^-$-N or $NH_4^+$-N or in the seasonal averages of $NO_3^-$-N, $NH_4^+$-N or PAN down to the 30 cm depth (Table 1).

**Table 1.** Seasonal averages for nitrate ($NO_3^-$-N) and ammonium ($NH_4^+$-N) at the 0–15 and 15–30 cm depths, and total plant available nitrogen (PAN) for 0–30 cm and associated standard errors in brackets (n = 32) for High fertilizer (High), Typical fertilizer (Typical), Late incorporation (Late), Organic and Control treatments.

| Treatment | $NO_3^-$-N kg ha$^{-1}$ (0–15 cm) | $NO_3^-$-N kg ha$^{-1}$ (15–30 cm) | $NH_4^+$-N kg ha$^{-1}$ (0–15 cm) | $NH_4^+$-N kg ha$^{-1}$ (15–30 cm) | Total PAN kg ha$^{-1}$ (0–30 cm) |
|---|---|---|---|---|---|
| High | 21.5 (3.9) [a] | 3.1 (0.5) [a] | 43.3 (6.7) [a] | 3.7 (1.5) [a] | 71.3 (1.8) [a] |
| Typical | 13.8 (2.4) [ab] | 1.9 (0.5) [ab] | 26.4 (4.5) [ab] | 1.4 (0.26) [ab] | 43.6 (3.7) [b] |
| Late | 12.1 (2.3) [b] | 2.5 (0.5) [ab] | 19.8 (6.8) [b] | 1.6 (0.5) [ab] | 36 (6.4) [b] |
| Organic | 8.8 (0.9) [b] | 2.1 (0.4) [ab] | 4.7 (0.6) [c] | 0.9 (0.1) [b] | 16.4 (0.8) [c] |
| Control | 8.6 (1) [b] | 1.8 (0.3) [ab] | 2.3(0.3) [c] | 0.7 (0.1) [b] | 13.4 (1.2) [c] |

**Note: Different letters indicate significant differences ($p < 0.05$) within the column.**

The overall seasonal average PAN supply was the greatest in the high treatment than all other treatments (Table 1). The Typical and Late treatments had 3.2 and 2.7 times more seasonal PAN than the control. The High treatment also supplied a significantly greater ($p < 0.05$) seasonal average $NH_4^+$-N and $NO_3^-$-N compared to the Late, Organic and Control treatments down to 30 cm depth and significantly more ($p < 0.05$) $NH_4^+$-N at 15–30 cm depth than the typical treatment. Similar to the observations of the seasonal supply of PAN, Typical and Late treatments supplied similar amounts of seasonal average $NO_3^-$-N and $NH_4^+$-N, but they also supplied similar amounts of $NO_3^-$-N and $NH_4^+$-N to the Organic and Control treatments. The only exception was $NH_4^+$-N at 0–15 cm depth, where Typical and Late treatments supplied four and six times more ($p < 0.05$) $NH_4^+$-N than Organic and Control treatments, respectively. Significant ($p < 0.05$) negative correlations were found between soil water content and $NO_3^-$-N (r = −0.47) and $NH_4^+$-N (r = −0.44). There was also a significant positive correlation ($p < 0.05$; $r = 0.45$) between average air temperature and $NO_3^-$-N (Figure A3).

Cumulative $NO_3^-$-N supply (Figure 5A) showed no significant differences among treatments ($p < 0.05$) until the middle of July 2016. At the final sampling time, the High treatment had three times more ($p < 0.05$) cumulative $NO_3^-$-N than Organic and Control treatments. There were no differences in cumulative $NO_3^-$-N supply between the Organic and Control treatments and also no significant differences between the High and Typical treatments. Cumulative $NO_3^-$-N was not different between the High and Late treatments. Cumulative $NH_4^+$-N supply (Figure 5B) increased steadily beginning at the time of GLSA biomass incorporation until the end of the growing season. The final sampling time showed that High treatments supplied significantly more ($p < 0.05$) than all other treatments, except for the Typical, and supplied six times more than the Control, whereas the Typical treatment supplied three times more than the Control. There were no differences between the Late and Organic treatments and the Control in total cumulative $NH_4^+$-N.

Potentially mineralizable nitrogen was dominated by $NO_3^-$-N, as after 2 weeks all $NH_4^+$-N was nitrified. Mineralizable $NO_3^-$-N displayed no significant differences during the initial 2-week incubation (Figure 6A). In the second incubation, the High treatment mineralized on average 4.3 kg ha$^{-1}$ day$^{-1}$ of $NO_3^-$-N, significantly ($p < 0.05$) more than all other treatments (Figure 6B) except for the Typical treatment. During the fourth and final incubation (Figure 6D), the High treatment on average mineralized 2.7 kg ha$^{-1}$ day$^{-1}$ of $NO_3^-$-N, significantly ($p < 0.05$) more than the Organic treatment and the Control, but this was not different from the Typical or Late treatments.

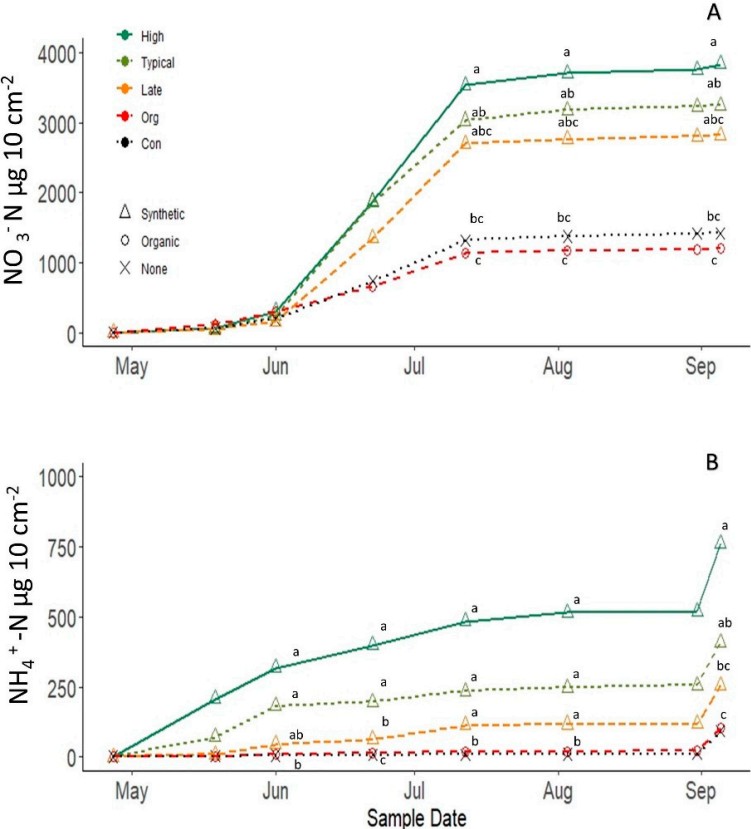

**Figure 5.** Cumulative nitrogen supply for High fertilizer (High), Typical fertilizer (Typical), Late incorporation (Late), Organic (Org) and Control (Con) treatments. Cumulative nitrate supply ($NO_3^-$-N) (**A**) and ammonium supply ($NH_4^+$-N) (**B**) during 2016 growing season. Triangles symbols indicate synthetic fertilizer, open circles are organic fertilizers and X symbol no amendments. Results are means of four replicates and different letters indicate significant differences ($p < 0.05$). Note that Y axes scale differs between figure (**A**) and (**B**).

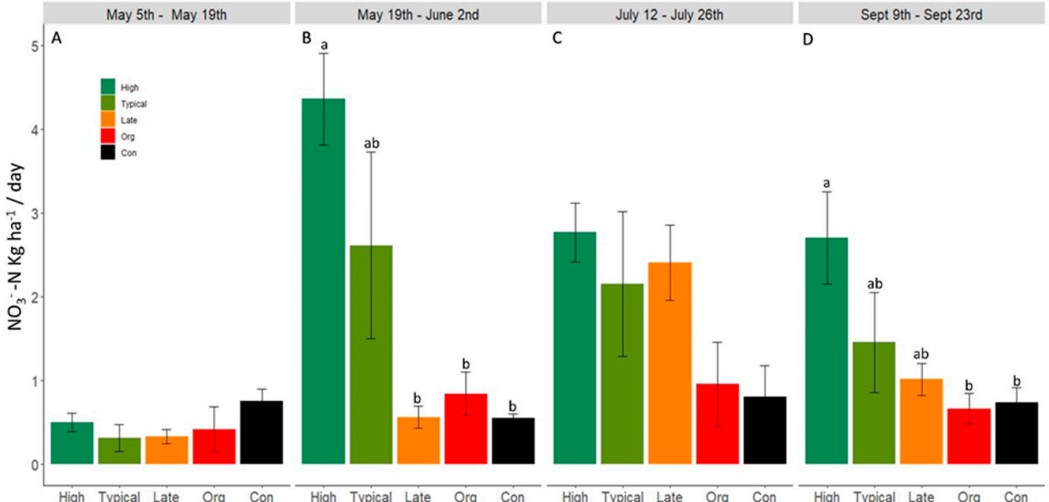

**Figure 6.** Potentially mineralizable nitrogen for High fertilizer (High), Typical fertilizer (Typical), Late incorporation (Late), Organic and control shown as mineralized ($NO_3^-$-N) during 5th May to 9th May (**A**); 19th May to 2nd June (**B**); from 12th July to 26th July (**C**); and from 9th September to 23rd September (**D**) incubation periods. Different letters indicate significant differences ($p < 0.05$).

## 4. Discussion

*4.1. On-Farm Regional Analysis of 3-Year-Old Grassland Set-Asides Shows Inconsistent Results*

Despite observing no differences in baseline soil properties (Table A1), specifically TC or TN, between fields in ACR and fields transitioned to cropping after three years of GLSA, significant treatment effects, albeit contrasting, were observed for both $NO_3^--N$ and $NH_4^+-N$ in the 2015 and 2016 growing seasons. In both seasons, $NO_3^--N$ availability in the GLSA did not peak until several weeks after the ACR, whereas $NH_4^+-N$ availability peaked before the ACR in 2015 and at the same time in 2016. In 2015, GLSA supplied on average 13 kg ha$^{-1}$ more $NO_3^--N$ and 5 kg ha$^{-1}$ more $NH_4^+-N$ down to 30 cm depth. Ion exchange probes, which tracked cumulative PAN supply, corroborated seasonal trends, with a higher PAN supply from GLSA than ACR in 2015, and less in 2016. These results show that although PAN mineralization may peak later in the season in GLSA compared to the ACR, the GLSA can still contribute to a higher seasonal average PAN, or GLSA may indeed reduce PAN relative to the ACR.

These confounding results are inconsistent with other GLSA studies that found PAN benefits for crop production following GLSA incorporation. For example, Chalmers et al. [2] followed PAN on fields after incorporation of 3-year perennial rye grass and perennial grass-white clover mixed GLSA on several soil types in England and found PAN benefits 30 kg ha$^{-1}$ down to 90 cm depth early in the production season of compared to annually cropped fields. Another study on a sandy loam soil in Ghent, Belgium tracked PAN in the first growing season following spring ploughing of a 3-year-old grazed GLSA and found 124–150 kg N ha$^{-1}$ more compared to continuously cropped fields [8].

One explanation for the limited PAN benefits observed in this study, could be due to the dominance of grasses in GLSA mix and high C:N ratio of the GLSA biomass, compared to other studies which contained higher proportion of legumes [2,34]. The C:N ratios observed in this study (Table A2) in both years was higher than 25:1 and probably resulted in some N immobilization. A GLSA study by Torstensson in Sweden found C:N ratios of 12 to 25 with different plant species compositions, with the lowest ratios containing white clover (*Trifolium repens* L.) and the highest with Timothy grass and meadow fescue (*Festuca pratensis* Huds.) [35]. The same study also found a considerably higher PAN in the soil profile after the incorporation GLSA biomass comprised exclusively of white clover compared to a biomass with white clover and grasses. On the other hand, Davies et al. [36] found that pure grass biomass of GLSA in Scotland mineralized an additional 205 kg N ha$^{-1}$ in the first year after the incorporation compared to grass-clover GLSA, which was attributed to a higher overall concentration of N in the grass residue compared to the grass clover primarily containing ryegrass roots deficient in N [36].

The percentage of clover in the GLSA biomass of this study was <1% of the total composition. The lack of clover survival after three years in GLSA was expected as it comprised only 2%, by seed weight, of the original seed mixture [4], and GLSA fields were not cut or grazed as is typical for fields in the lower Fraser River delta. Set-aside management, such as grazing tends to benefit the spread of clover and could potentially help maintain or even increase clover proportion in the GLSA [37]. Increasing the percentage of legume seeds in the GLSA planting mix or allowing grazing could enhance mineralization [38]. The higher proportion of grasses and thus higher C:N ratio of the GLSA biomass in 2015 may help to explain why there was a delayed mineralization and overall greater cumulative PAN in 2015 than in 2016.

*4.2. The Impact of the Timing of GLSA Incorporation, Fertilizer Rate and Type on N-Dynamics*

The inconsistent performance of GLSA over the two growing seasons could also be explained by the difference in timing of GLSA incorporation between each season. In 2015, GLSA incorporation was delayed to mid-May due to considerable early season rainfall, while in 2016, the GLSA was incorporated in April, due to a drier spring than in 2015 and thus the earlier workability of fields. It is possible that earlier incorporation in 2016 meant cooler soils and thus lower mineralization

rates than 2015. The findings of the controlled field-scale study indicate that delaying incorporation of the GLSA can increase (as we hypothesized) the mineralization rate (although we did not see any significant differences in PMN between the Typical and Late timing). Thus, despite starting two weeks later than the Typical treatment, the Late treatment (with an equivalent amount of fertilizer) achieved the same overall seasonal average and cumulative PAN. The significant negative correlation of both $NO_3^-$-N and $NH_4^+$-N with soil water content and positive correlation of $NO_3^-$-N with air temperature indicate that the cool, wet spring in the Fraser River delta controls the timing of N supply following the incorporation of GLSA biomass. These indicators are limiting in that soil moisture and temperature were measured at the time of soil sampling, while plot-specific soil and climate information or other indicators (e.g., hydrothermal coefficients) integrated over time may have resulted in different outcomes.

While we had hypothesized that higher fertilizer application rates would result in earlier and greater PAN release, our findings show only an increase in average season PAN in the Typical treatment. Both the High and Typical fertilizer treatments appeared to release PAN with a similar magnitude and timing to what we observed in fields across the region following GLSA. At both $NH_4^+$-N peaks, the Typical treatment supplied 45 kg ha$^{-1}$ more compared to the GLSA fields in the regional analysis. Soil $NO_3^-$-N showed similar patterns, with High and Typical treatments supplying 65 kg ha$^{-1}$ and Typical 40 kg ha$^{-1}$ at 56 DAI compared to regional GLSA fields at 38 kg ha$^{-1}$. Comparing cumulative PAN, however, showed that High and Typical treatments supplied substantially more PAN at the end of season compared to regional GLSA estimates, with the High treatment supplying eight times more cumulative $NO_3^-$-N and 3.7 times more cumulative $NH_4^+$-N, and the Typical treatment supplied three times more $NO_3^-$-N and four times more $NH_4^+$-N. Given no differences in timing but greater cumulative $NO_3^-$-N, it is likely the High application rate would leave higher residual soil N. Unfortunately, we did not have a treatment with only fertilizer in the controlled experiment to match that of the ACR fields; hence, we cannot report what the contribution of the interaction between the fertilizer and GLSA was relative to fertilizer alone.

Although plant uptake could explain some of the difference between the regional values and the controlled plot-scale experiment, it is also likely that the fields under organic production included in the regional study lowered the mean PAN values. The controlled plot-scale study clearly highlighted the importance of fertilizer type (i.e., organic or synthetic) as a factor in determining PAN following GLSA. In this study, treatments with synthetic N inputs maintained high PAN levels throughout most of the growing season, while the Organic treatment was not significantly different than the Control. Mineralization rates from organic amendments can be quite variable [39] and are known to typically delay N release compared to readily available synthetic fertilizers [40]. It is also possible that the calculated mineralization rate used for this study (15%) underestimated the amount required to reach 100 kg of PAN within the plot, which could explain the drastic undersupply of PAN from the Organic treatment. Without having an organic fertilizer only treatment, it is impossible to determine how much of this was due to the interaction with the GLSA biomass.

*4.3. Implications for the Farming System*

Synchronizing N availability and crop demand is imperative to maximizing crop benefits and minimizing detrimental impacts to the environment when transitioning GLSA fields back into crop production. If synchronized appropriately to crop demand, PAN supplied from GLSA will be utilized efficiently to increase crop yields, whilst also minimizing the need for additional fertilizer applications [41]. In this study, the lack of significant differences in relative crop yield between GLSA and ACR indicate that the differences observed in PAN were not substantial enough to impact the crop. Alternatively, the variability in management across the fields could also explain the high variability in relative crop yields and a lack of significant differences in PAN results. Other studies have found that GLSA have positive impacts on subsequent crop yields. For example, a study by Nevens and Reheul reported an 85% increase in silage corn grown following incorporation of 3-year-old grazed GLSA,

relative to a control with no additional fertilizer application [8]. Given the marginal PAN increase observed in 2015 in this study and the lack of PAN increase or differences in crop yields, the benefits of GLSA for subsequent cropping remain negligible in this system, but warrant further investigation.

On one hand, these results may indicate that the 3-year-GLSA does not provide a large PAN benefit, but also indicate that the farmers in the GLSA program are not over supplying N. If N release is delayed by immobilization and if not taken up by the crop, it will remain in the soil typically in the form $NO_3^--N$, which is known to be highly susceptible to leaching into the ground water [39–41]. Increased residual soil nitrogen (RSN) at the end of the production season may be a clear environmental tradeoff in GLSA rotations that increase PAN during the season as additional N benefits to farmers, if not utilized by the crop during the growing season, could result in $NO_3^--N$ leaching and lead to negative environmental impacts. In our study, GLSA fields in 2015 left only an average of 13 kg $NO_3^--N$ ha$^{-1}$ more residual soil N in the 0–30 cm depth compared to ACR, and in 2016 there were no significant differences. Other studies have found GLSA fields to increase the likelihood of ground water contamination, especially in the first year back in a crop production. For example, a study on 3-year-old grass and clover GLSA on sandy soils in Denmark observed large amounts of N leaching (63–216 kg N ha$^{-1}$) in the first year after spring ploughing [42], while a study at the Rothamsted Experimental station with lower precipitation than the Fraser River delta observed losses estimated at 118 kg N ha$^{-1}$ in the first winter after incorporation of a 3-year-old rye-grass-clover GLSA [43]. In our study, a much lower leaching potential for GLSA and ACR fields alike going into the winter was observed in both years, far below the 100 kg ha$^{-1}$ that has been identified as a provincial general threshold of concern (regardless of soil type) by the BC Ministry of Agriculture [44]. These low RSN could be a result of the low clover content in these set-asides and efficient use of N by the crop. Further studies should attempt to track mineralized N over the winter season to better evaluate the environmental impact of GLSA.

The findings of the regional and controlled plot-scale study do illustrate some of the management decisions that likely affect the amount of PAN a GLSA could provide relative to continuously cropped fields. Farmers in the Fraser River delta region of BC should expect a shift in the timing of PAN, particularly if they are incorporating their GLSA later in the season. Farmers may choose to initially apply less fertilizer in expectation of N benefits or if planting an early crop may forego additional fertilizer banding in the season in expectation of this subsequent N peak. Either way, this N benefit from GLSAs represents an opportunity to reduce production costs from additional fertilizer inputs, but the farmer's selection of planting times will largely determine how much of a benefit is captured. Furthermore, the controlled plot-scale study highlights the need for additional care when using GLSA as a transition method to an organic farming system. The variable N supply from organic fertilizers and the high C inputs from GLSA may cause additional N immobilization, and as shown in this study, could cause low N supply in the first season following GLSA cessation, as seen in our 2016 results. Thus, accurate estimates of GLSA N credits would help to ensure adequate cash crop yield and minimize potential environmental impacts from RSN.

## 5. Conclusions

Through a regional analysis carried out in 2015 and 2016, we quantified the impacts of 3-years GLSA on PAN in the first growing season following GLSA incorporation as well as crop yields and residual soil N. We found that a substantial amount of N was added to GLSA fields through the incorporation of AGB ranging from 86 to 101 kg ha$^{-1}$ over both seasons. In 2015, GLSA contributed an additional seasonal average of 18 kg N ha$^{-1}$ relative to ACR fields, but the opposite was true in 2016, when ACR fields supplied 20 kg N ha$^{-1}$ more compared to GLSA at the 0–30 cm depth. In both years, PAN following GLSA peaked later in the season than in the ACR, likely due to immobilization of N facilitated by the incorporation of biomass with high C:N ratio. Although we found, as expected, higher residual soil N in the GLSA fields in 2015, this did not hold true in 2016. There were no differences in relative crop yields between GLSA and ACR.

The results of our controlled plot-scale experiment corroborate much of what we observed in the on-farm study and illustrated the effects of various management practices used in conjunction with GLSA on PAN following the incorporation. The findings suggest that the doubling of fertilizer applications from Typical to High did not reduce the immobilization of N or supply more PAN earlier in the season, suggesting that farmers should be able to apply typical rates without the worry of additional early season immobilization from the GLSA. Delaying the incorporation of GLSA did not result in reduced cumulative PAN and affirms findings that the later incorporation in the regional analysis resulted in greater PAN benefits due to higher temperatures and more favorable soil moisture. The controlled plot-scale experiment also very clearly demonstrated limited mineralization from GLSA in combination with organic fertilizers. The results from this study suggest that a 3-year-old GLSA does not result in consistent PAN benefits or improved crop yields, but our study was limited by the short duration and relatively small sample size. Although this study provides insight into potential management decisions which could be taken by farmers following participation in the GLSA program, these results should be corroborated by subsequent studies to confirm and further elucidate the impacts of GLSA cessation on subsequent N availability and impacts on cash crop yield and quality.

**Author Contributions:** Conceptualization, K.W. and S.M.S.; methodology, K.W. and S.M.S.; software, K.W.; validation, K.W., S.M.S. and M.K.; formal analysis, K.W. and S.M.S.; investigation, K.W.; resources, S.M.S., M.K. and D.B.; data curation, K.W. and S.M.S.; writing—original draft preparation, K.W., S.M.S., and M.K.; writing—review and editing, K.W., S.M.S., M.K., and D.B.; visualization, K.W., S.M.S., and M.K.; supervision, S.M.S. and M.K.; project administration, S.M.S., D.B., and M.K.; funding acquisition, S.S., M.K., and D.B. All authors have read and agreed to the published version of the manuscript.

**Funding:** Federal funding for the project was delivered by the Investment Agriculture Foundation of BC. Additional funding was provided by Mitacs, and the DF&WT.

**Acknowledgments:** We are grateful to the farmers in Delta, BC for allowing us to establish experiments on their fields, the Delta Farmers Institute for organizing outreach, as well as Christine Terpsma and all undergraduate research assistants and volunteers who helped with field sampling and laboratory work.

**Conflicts of Interest:** The authors declare no conflict of interest.

**Appendix A**

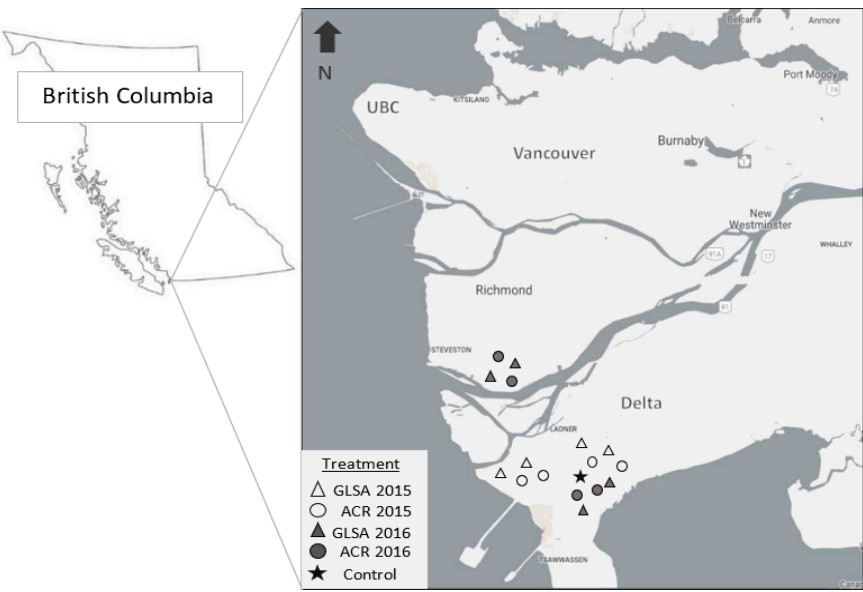

**Figure A1.** Map of the Lower Fraser River delta with the location of regional analysis fields in annual crop rotations (ACR-circles) and following 3-year grassland set-asides (GLSA-triangles) in 2015 and 2016 as well as the controlled experiment (star) in Delta, British Columbia, Canada.

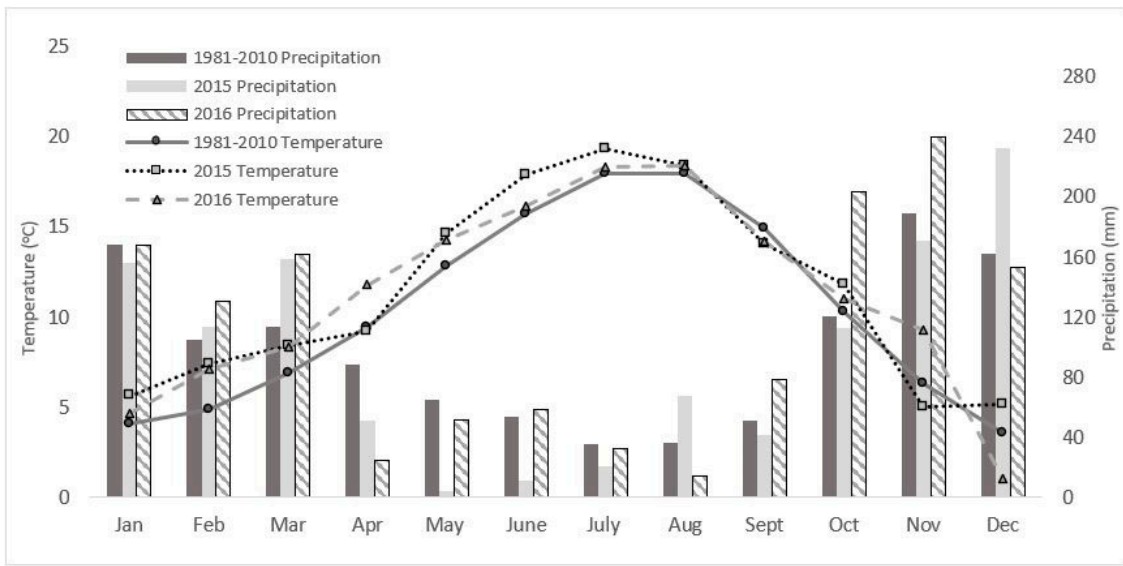

**Figure A2.** Total monthly precipitation and mean monthly temperature data from Vancouver International airport weather station [18].

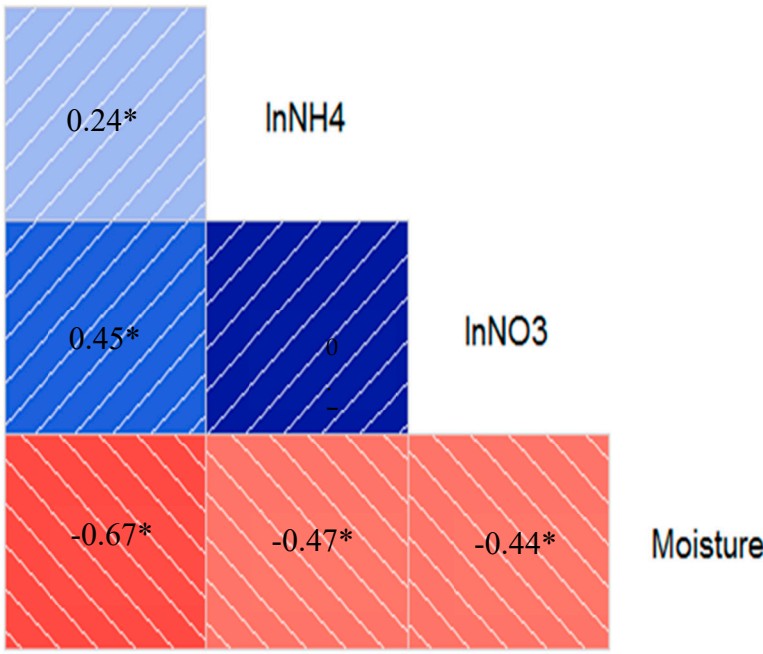

**Figure A3.** Pearson's correlation matrix between measured PAN and climatic variables. All variables were transformed with a log base 10. Variables included were average temperature (Average.Temp), soil moisture (Soil.fracwet), nitrate (lnNO3), and ammonium (lnNH4). Values represent positive (blue) or negative (red) correlation coefficients (r) with darker shading representing stronger correlations and values with stars (*) represent significant correlations (*p* < 0.05).

**Table A1.** Above ground biomass (AGB), carbon to nitrogen (C:N) ratio, and additional nitrogen (AN) means and associated standard errors (n = 4) of the grassland set-aside in the 2015 and 2016 growing seasons.

| Year | AGB (kg ha⁻¹) | C:N Ratio | AN kg ha⁻¹ |
|------|---------------|-----------|------------|
| 2015 | 6024 (637) | 31 (2.3)[a] | 86.4 (12.5) |
| 2016 | 5709 (345) | 25 (2.2)[b] | 101.4 (6.6) |

Note: Different letters indicate significant differences ($p < 0.05$).

**Table A2.** Total soil carbon (TC), total soil nitrogen (TN), pH, electrical conductivity (EC), sand, silt, and clay at 0–15 cm depth taken at the beginning of 2015 and 2016 seasons. Statistical analyses were run but no differences were found ($p < 0.05$) between grassland set-aside (GLSA) and annual crop rotation (ACR) for any of these soil properties.

| Year | Treatment | TC (g C kg⁻¹) | TN (g N kg⁻¹) | pH | EC (ds m⁻¹) | Sand (%) | Silt (%) | Clay (%) |
|------|-----------|---------------|---------------|-----|-------------|----------|----------|----------|
| 2015 | GLSA | 25.43 (3.26) | 2.25 (0.23) | 5.82 (0.04) | 2.7 (0.6) | 7 (0.7) | 66 (0.8) | 28 (0.9) |
|      | ACR | 24.57 (4.32) | 2.12 (0.28) | 5.85 (0.04) | 2.9 (1.6) | 9 (1.9) | 64 (1.2) | 27 (1) |
| 2016 | GLSA | 27.27 (5.09) | 2.3 (0.39) | 5.29 (0.34) | 1.0 (0.2) | 8 (0.6) | 66 (0.7) | 28 (0.9) |
|      | ACR | 29.05 (4.43) | 2.58 (0.39) | 5.58 (0.2) | 1.2 (0.3) | 7 (0.9) | 64 (0.6) | 29 (0.6) |

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
