# Peer review of "Nitrogen Dynamics Following Incorporation of 3-Year Old Grassland Set-Asides in the Fraser River Delta of British Columbia"

_agronomy, doi:10.3390/agronomy10091382_

Round 1

Reviewer 1 Report

I have reviewed the manuscript entitled “Nitrogen dynamics following incorporation of 3-year old grassland set-asides in the Fraser River delta of British Columbia”. The study aims to evaluate the potential impact of short-term grassland Set-aside (GLSA) on N availability of the subsequent cash-crop. I really like the study because it comprises two different field studies which together provide a   valuable information about the N dynamics after GLSA incorporation. The data obtained are very valuable and the conclusions obtained from the dataset are very interesting for farm managers and policy makers. Therefore, I consider that the manuscript could be suitable for publication in “Agronomy” journal after few considerations.

The main weakness of the study in the introduction. I consider it is very short and it is not presenting well the study. I consider the authors should enlarge more the part where the problems of N immobilization are presented (L51-57). Then, in the manuscript, we found that the application of N fertilizer or the late incorporation could mitigate the potential N immobilization after GLSA incorporation. However, that is not mentioned in the introduction, only in the hypothesis section (which were surprising because these topics were not presented previously).  How important is the timing? How important is the application of fertilizer? I think the authors should provide more information about alternative GLSA management in order to introduce the hypothesis. In addition, I think that a brief description of the two complementary studies would help to understand the hypothesis.

The section 2.1 should be divided in two, one for each complementary study, in order to clarify.

I had a doubt when I was reading the manuscript. Why did the authors measured PAN by mean of NH4 and NO3 determination and by Ion exchange membranes? I have not found any explanation in the manuscript about the differences between both approaches.

In the statistical analysis section, the correlations from Figure S3 were nor described.

The units of TC and TN should be changed to SI units (mg kg-1)

The result section seems to be too long and I have the feeling that no all the explained results are used in the discussion. Maybe the authors should focus on the bigger picture and delete these results which are not discussed. The authors could look in the figures for the details.

I think that the low proportion of legumes in the swards was affecting the results as pointed out by the authors (l397-405). I only have a question, the low proportion of legumes in the seed mixture was selected on purpose or is the common seed mixture in the area? The use of a seed mixtures with higher legume proportion probably would increase the PAN differences between treatments. However, in general, I don’t agree with those studies where the legume proportion in the swards is increased to 40-60% of biomass because the obtained results are very unrealistic (maintain a legume proportion of 50% in swards requires a hard grassland management). Therefore, I would appreciate if the authors can clarify why the seed mixture had a low legume proportion and explain if under the local condition, that mixture is common. That would increase the applicability of the obtained results.

The authors did not discuss the advantages and disadvantages of ammonium and nitrates in soil. For example, nitrates are easily used by plant but the risk of leaching or denitrification is higher than in ammonium form. Therefore, if the N is accumulated in ammonium form, the risk of N losses will be lower than if the N is accumulated in nitrate form. In addition, during the nitrification process there is N2O production.

In general, I think the authors should consider the higher nitrate production as a risk instead as an advantage. For example, the higher accumulation of nitrates in September for “high” treatment can be interpreted as a risk of N loss and negative environmental impact. If the crop yield of a treatment is similar but the nitrate production is lower, it could be considered as an advantage. A further evaluation of the treatments considering the potential differences in N losses should be done. 

L53: Please, present the N abbreviation   

L90: DF&WT mix?

L139: -55°?

L159: Ratio of soil:extractant?

L203: and normality.

L393: complete the reference

L407: Please, do not start a new section using “another explanation”

Author Response

Dear Reviewer 1,

We appreciate your detailed and constructive review of our manuscript.  We have responded to each of your points below in and have tracked the changes we have made in the attached Word document.

Reviewer 1

I have reviewed the manuscript entitled “Nitrogen dynamics following incorporation of 3-year old grassland set-asides in the Fraser River delta of British Columbia”. The study aims to evaluate the potential impact of short-term grassland Set-aside (GLSA) on N availability of the subsequent cash-crop. I really like the study because it comprises two different field studies which together provide a   valuable information about the N dynamics after GLSA incorporation. The data obtained are very valuable and the conclusions obtained from the dataset are very interesting for farm managers and policy makers. Therefore, I consider that the manuscript could be suitable for publication in “Agronomy” journal after few considerations.

  1. The main weakness of the study in the introduction. I consider it is very short and it is not presenting well the study. I consider the authors should enlarge more the part where the problems of N immobilization are presented (L51-57). Then, in the manuscript, we found that the application of N fertilizer or the late incorporation could mitigate the potential N immobilization after GLSA incorporation. However, that is not mentioned in the introduction, only in the hypothesis section (which were surprising because these topics were not presented previously).  How important is the timing? How important is the application of fertilizer? I think the authors should provide more information about alternative GLSA management in order to introduce the hypothesis.

Response to point 1: We appreciate that in our effort to be concise we neglected some important background information that would help set the stage for why we did this study.  As the reviewer suggested, we have expanded the introduction substantially LN55-84 to provide more details of the logic behind research objectives and hypotheses.

  1. In addition, I think that a brief description of the two complementary studies would help to understand the hypothesis.

Response to point 2: The following text was added to manuscript in L87-90 to help frame the two studies:

“To do so, we conducted an observational experiment to track PAN following GLSA incorporation under a range of production conditions across a number of operational farms over two seasons, and a complementary controlled plot experiment where variation in conditions were minimized over one season.”

  1. The section 2.1 should be divided in two, one for each complementary study, in order to clarify.

Response to point 3: In response to the comment Section 2.1 was divided into two sections; On- Farm Regional Assessment and Controlled Plot-Scale Experiment

  1. I had a doubt when I was reading the manuscript. Why did the authors measured PAN by mean of NH4 and NO3 determination and by Ion exchange membranes? I have not found any explanation in the manuscript about the differences between both approaches.

Response to point 4: We have added details in Section 2 LN17-181 to explain the differences in methods.

“While these soil samples were important for quantifying area-based values of PAN, they only provided a snapshot in time.  To ensure that we did not miss important mineralization events and compare cumulative PAN, we also deployed ion exchange resin probes.  Ion exchange membranes were used over the growing season at 2- to 4-week intervals. Probes with a surface area of 10 cm2, made from ion resin membranes which adsorb cations and anions, were installed vertically into the top 15 cm of the soil profile. Two probe pairs were placed in each subplot.”

  1. In the statistical analysis section, the correlations from Figure S3 were nor described.

Response to point 5: The following text was added to manuscript in LN281-282 to address the comment.

“A correlation matrix was run using Pearson’s correlation and utilizing the Corrgram package [29] between various soil properties, sampled and climate variables.”

  1. The units of TC and TN should be changed to SI units (mg kg-1)

Response to point 6:  We have converted the % values to g kg-1 soil

  1. The result section seems to be too long and I have the feeling that no all the explained results are used in the discussion. Maybe the authors should focus on the bigger picture and delete these results which are not discussed. The authors could look in the figures for the details.

Response to point 7:  We agree that we were trying to present too much information in this section. In the attempt to shorten this section, we have moved tables 1 and 2 to the supplemental materials and deleted the description of those results as they are not directly relevant to our hypothesis.

  1. I think that the low proportion of legumes in the swards was affecting the results as pointed out by the authors (l397-405). I only have a question, the low proportion of legumes in the seed mixture was selected on purpose or is the common seed mixture in the area? The use of a seed mixtures with higher legume proportion probably would increase the PAN differences between treatments. However, in general, I don’t agree with those studies where the legume proportion in the swards is increased to 40-60% of biomass because the obtained results are very unrealistic (maintain a legume proportion of 50% in swards requires a hard grassland management). Therefore, I would appreciate if the authors can clarify why the seed mixture had a low legume proportion and explain if under the local condition, that mixture is common. That would increase the applicability of the obtained results.

Response to point 8:  The seed mix is the commonly applied and recommended seed mixture by the set-aside implementing organization - the Delta Farmland and Wildlife Trust. The seed mixture proportions were determined in order to achieve the following objectives:

  • provide low maintenance requirements
  • enhance soil organic matter and structure
  • establish good vegetative cover for foraging, roosting, and nesting wildlife
  • be relatively inexpensive, and
  • very importantly to resist browsing by the large populations of migrating waterfowl that visit the region each winter.

Furthermore, the GLSA seed mix was selected through previous studies that determined the type of plant species and their proportions to address all the objectives listed above. It is essential that the GLSA seed mix provide habitat for small mammals to support birds of prey and to enhance soil organic carbon and in turn soil structure, which are the common forms of degradation in this region. Many soils in this region are saline due to their proximity to the ocean, hence, plant species in the GLSA seed mix needed to be capable to withstand moderate salinity issues. Lastly, the GLSA seed mix also needed to provide a for a short-term income to farmers through haying.

Thus, due to low legume proportion in the GLSA seed mix and subsequent low PAN following GLSA cessation are expected. Any potential changes to the GLSA seed mix in order to achieve subsequent PAN benefits need to be weighed against the initial objectives of the GLSA stewardship program. We have added to following to LN122-127 to clarify this briefly:

“The species and proportions which comprise the seed mix were selected through previous studies with the goal to improve soil degradation, commonly caused by the intensive tillage practices used for potato production in this region [19]. The GLSA mix is also comprised of plant species that are salt tolerant and resist browsing by the large populations of migrating waterfowl that visit the region every year while meeting the other objectives of the DF&WT program listed above.”

  1. The authors did not discuss the advantages and disadvantages of ammonium and nitrates in soil. For example, nitrates are easily used by plant but the risk of leaching or denitrification is higher than in ammonium form. Therefore, if the N is accumulated in ammonium form, the risk of N losses will be lower than if the N is accumulated in nitrate form. In addition, during the nitrification process there is N2O production. In general, I think the authors should consider the higher nitrate production as a risk instead as an advantage. For example, the higher accumulation of nitrates in September for “high” treatment can be interpreted as a risk of N loss and negative environmental impact. If the crop yield of a treatment is similar but the nitrate production is lower, it could be considered as an advantage. A further evaluation of the treatments considering the potential differences in N losses should be done. 

The following text was added to manuscript in LN553-559to address the comment.

“I If N release is delayed by immobilization and if not taken up by the crop, it will remain in the soil typically in the form NO3--N, which is known to be highly susceptible to leaching into the ground water [39–41]. Increased residual soil nitrogen (RSN) at the end of the production season may be a clear environmental tradeoff in GLSA rotations that increase PAN during the season as additional N benefits to farmers, if not utilized by the crop during the growing season, could result in NO3--N leaching and lead to negative environmental impacts.

And to L570-572

“These low RSN could be a result of the low clover content in these set-asides and efficient use of N by the crop. Further studies should attempt to track mineralized N over the winter season to better evaluate the environmental impact of GLSA.”

  1. L53: Please, present the N abbreviation   

Response to point 10:  We have added “nitrogen (N)” at the first use in both the abstract and in L55

  1. L90: DF&WT mix?

Response to point 10:  We have clarified that “The GLSA fields were seeded with a mixture of grass and legumes developed by the DF&WT “ LN119-120.

  1. L139: -55°?

Response to point 12: Samples were freeze dried at -55 degree Celsius (-55oC)

  1. L159: Ratio of soil:extractant?

Response to point 13: We have clarified that the ratio was 10 grams of soil to 25 ml of 2M KCL.

  1. L203: and normality.

Response to point 14: Added to LN278

  1. L393: complete the reference

Response to point 15: The full reference has been added to manuscript and to the reference section.

  1. L407: Please, do not start a new section using “another explanation”

Response to point 16: The following text was added to manuscript in LN489-490 to address the comment.

Sentence changed to “The inconsistent performance of GLSA over the two growing seasons could also be explained by the difference in timing of GLSA incorporation between each season.”

Reviewer 2 Report

The work is an interesting contribution to a better understanding of soil mechanisms due to GLSA incorporation, regardless of the fact that I believe that these are short-term studies and no general conclusions can be drawn from them. In my opinion, important and valuable of this work is the combination of research in production conditions with experimental field (plot) research. I appreciate both the research hypotheses and the methods of their verification (however, I did find some space for improvement in the design of the experiment). Comprehensive, clear and varied in form presentation of the results (including statistics) are the strong point of the study. Both the discussion and the summary are accurate and strictly relate to the obtained and presented results.
Summing up, I emphasize that the results presented in the paper should be verified by continuation of the research, because balance studies should be conducted in a longer time perspective.

Author Response

Dear Reviewer 2,

We appreciate your detailed and constructive review of our manuscript.  We have responded to each of your points below in and have tracked the changes we have made in the attached word document.

Reviewer 2

Comments and Suggestions for Authors

The work is an interesting contribution to a better understanding of soil mechanisms due to GLSA incorporation, regardless of the fact that I believe that these are short-term studies and no general conclusions can be drawn from them. In my opinion, important and valuable of this work is the combination of research in production conditions with experimental field (plot) research. I appreciate both the research hypotheses and the methods of their verification (however, I did find some space for improvement in the design of the experiment).Comprehensive, clear and varied in form presentation of the results (including statistics) are the strong point of the study. Both the discussion and the summary are accurate and strictly relate to the obtained and presented results.
Summing up, I emphasize that the results presented in the paper should be verified by continuation of the research, because balance studies should be conducted in a longer time perspective.

Feedback on attached PDF:

  1. L111: The treatment with fertilization and without AGB incorporation is missing in my opinion.

Response to point 1:  This is an important point and one that we would change if we could. Given that this cannot be done, we have added a caveat to the discussion illustrating the limitations of the study.  In LN501-504 we have added:

“These indicators are limiting in that soil moisture and temperature were measured at the time of soil sampling, while plot specific soil and climate information or other indicators (e.g. hydrothermal coefficients) integrated over time may have resulted in different outcomes.”

  1. L134: “Analysenysteme” – check spelling

Response to point 2:  Spelling changed to “Analysensysteme”

  1. L301-303 – here you can see the lack of a treatment with only mineral fertilization

Response to point 3: Yes, we agree, this is where it would have been helpful to have the 6th treatment.

  1. L364: “inconsistent” – check spelling

Response to point 4: Spelling changed to “Inconsistent”

  1. L406: “Fertilzer” – check spelling

Response to point 5: Spelling changed to “Fertilizer”

  1. L415-417: numerous comparisons to weather conditions suggest more detailed comparisons e.g. relations to hydrothermal coefficients for the decade or week in which the samples were taken.

Response to point 6:  This is an important limitation to our study.  We have added the following in LN501-504 to bring this to the attention of the reader:

“These indicators are limiting in that soil moisture and temperature were measured at the time of soil sampling, while plot specific soil and climate information or other indicators (e.g. hydrothermal coefficients) integrated over time may have resulted in different outcomes.”

  1. L467: here we talk about leaching potential and compare to lysimeter studies (possibly) in sandy soils. This may result in a difference in results, in addition pay attention to the level of fertilization and fertilizer type (slurry)

Response to point 7:  We have clarified that this was not a comparison to other studies but rather related to provincial legislation for nutrient management.  LN567-570 now reads:

“In our study, a much lower leaching potential for GLSA and ACR fields alike going into the winter was observed in both years, far below the 100 kg ha-1 that has been identified as a provincial general threshold of concern (regardless of soil type) by the BC Ministry of Agriculture [44]”

  1. L505: The results of this study are effectively a one-year study and definitely need to be continued.

Response to point 8:  We agree that the duration of this analysis is a limitation.  The following text was added to manuscript in LN615-620 to address the comment.

“Results from this study suggest that 3-year-old GLSA are not resulting in consistent PAN benefits or improved crop yields, but are limited by the short duration and relatively small sample size. Although this study provides insight into potential management decisions which could be taken by farmers following participation in the GLSA program, these results should be corroborated by subsequent studies to confirm and further elucidate the impacts of GLSA cessation on subsequent N availability and impacts on cash crop yield and quality.” 

Round 2

Reviewer 1 Report

I have reviewed again the manuscript entitled "Nitrogen dynamics following incorporation of 3-year old grassland set-asides in the Fraser River delta of British Columbia" and I'm very satisfied with the changes done by the authors. I think they solved properly all my suggestions and questions and now, the manuscript it's ready for publication.